# Cross-Species Transmission of Swine Hepatitis E Virus Genotype 3 to Rabbits

**DOI:** 10.3390/v12010053

**Published:** 2020-01-02

**Authors:** Sang-Hoon Han, Byung-Joo Park, Hee-Seop Ahn, Yong-Hyun Kim, Hyeon-Jeong Go, Joong-Bok Lee, Seung-Yong Park, Chang-Seon Song, Sang-Won Lee, Yang-Kyu Choi, In-Soo Choi

**Affiliations:** 1Department of Infectious Diseases, College of Veterinary Medicine, Konkuk University, 120 Neungdong-ro, Gwangjin-gu, Seoul 05029, Korea; hansh11@naver.com (S.-H.H.); twilightsd@naver.com (B.-J.P.); heesuob2@naver.com (H.-S.A.); yongkim22@naver.com (Y.-H.K.); misilseju@naver.com (H.-J.G.); virus@konkuk.ac.kr (J.-B.L.); paseyo@konkuk.ac.kr (S.-Y.P.); songcs@konkuk.ac.kr (C.-S.S.); odssey@konkuk.ac.kr (S.-W.L.); 2Department of Laboratory Animal Medicine, College of Veterinary Medicine, Konkuk University, 120 Neungdong-ro, Gwangjin-gu, Seoul 05029, Korea; yangkyuc@konkuk.ac.kr

**Keywords:** HEV, rabbit, swine, antibody, RNA, transmission, inflammation

## Abstract

Hepatitis E virus (HEV) is a quasi-enveloped, positive-sense single stranded RNA virus. HEV continually expands the host ranges across animal species. In this study, the possibility of cross-species infection with swine HEV-3 was investigated using rabbits. A total of fourteen 8-week old, specific pathogen-free rabbits were divided into three experimental groups. Four rabbits were used as negative controls, four rabbits were infected with rabbit HEV as positive controls, and six rabbits were inoculated with swine HEV-3. HEV RNA were detected from serum and fecal samples after viral challenge. The levels of anti-HEV antibodies, pro-inflammatory cytokines (IL-1, IL-6, TNF-α and IFN-α), and liver enzymes (alanine and aspartate aminotransferases) were determined in serum samples. Histopathological lesions were examined in liver tissues. Viral RNA and anti-HEV antibodies were identified in rabbits inoculated with swine HEV-3 demonstrating positive infectivity of the virus. However, pro-inflammatory cytokine and liver enzyme levels in serum were not significantly elevated, and only mild inflammatory lesions were detected in the liver tissues of rabbits infected with swine HEV-3. These results suggest that swine HEV-3 can engage in cross-species transmission to rabbits, but causes only mild inflammation of the liver.

## 1. Introduction

Hepatitis E virus (HEV), of the family *Hepeviridae*, is a quasi-enveloped, positive-sense single stranded RNA virus with an approximately 7.2 kb-long genome [1]. HEV infection is a major public health concern, worldwide. HEV infection usually causes acute hepatitis with self-limiting outcomes [2]. However, chronic hepatitis cases have been reported in immune-compromised individuals, such as human immunodeficiency virus-1 patients, solid organ transplant recipients, and patients with hematological malignancies who are infected with HEV-3 [3,4,5,6]. The mortality rate in HEV infection is relatively low (<1%), but can reach more than 20% in pregnant women who are infected with HEV-1 and HEV-2 in developing countries, such as India [7].

HEV, with only one serotype, has been classified into eight genotypes; HEV-1 to HEV-8 [1,8]. HEV-1 and HEV-2 only infect humans. However, HEV-3 and HEV-4 have been isolated from humans and several animal species including pigs [9,10]. HEV-3, HEV-4 and recently identified HEV-8, are recognized as zoonotic viruses that cause infections in humans through cross-species transmission from other animal species [11]. People who frequently come in contact with domestic and wildlife animals, such as veterinarians and famers, have a higher risk of HEV infection than other people [12,13]. This suggests that some animals play important roles as HEV reservoirs involved in the transmission of HEV to humans. Animal models have been developed to study the pathogenesis and cross-species transmission of HEV. Initially, pigs and macaques were used to verify the pathogenesis of HEV strains originating from humans and animals [14,15,16,17]. Recently, rabbits have also been recognized as hosts of HEV [18]. Furthermore, HEV isolated from rabbits could infect humans, pigs and macaques [19,20,21,22]. However, there are some limitations of using pigs or macaques as experimental animal models for HEV studies, such as high expense, handling difficulty, and ethical challenges. Therefore, new animal models for the study of cross-species transmission of HEV are needed. Rabbits have already been used for the study of pathogenesis and cross-species transmission of HEV [23,24]. However, a limitation of these previous studies is that studies were mainly focused on the pathogenesis of human strains of HEV. In this study, cross-species transmission of swine HEV-3 was attempted in rabbits. Detection of HEV RNA from both fecal and blood samples and anti-HEV antibodies in sera indicated that swine HEV was successfully transmitted to rabbits across the species barrier.

## 2. Materials and Methods

### 2.1. Experimental Animals and Viruses

All animal experiments were approved at 14 April, 2017 by the Institutional Animal Care and Use Committee of Konkuk University (IACUC No. KU16128). Rabbits were kept in the animal facility of Konkuk Laboratory Animal Research Center. A total of 14 8-week-old specific pathogen-free New Zealand white rabbits were used for this experiment (Samtako, Gyeonggi, Korea). Rabbits were divided into three experimental groups: a negative control with mock infection (*n* = 4), a positive control group infected with rabbit HEV (*n* = 4), and a cross-species transmission group infected with swine HEV-3 (*n* = 6). Rabbits were intravenously infected with rabbit HEV and swine HEV-3. Rabbit HEV (GenBank No. KY496215) and swine HEV-3 (GenBank No. MF095679) used in this study were obtained from the stool sample of a rabbit and a pig, respectively, as described in our previous studies [25,26]. The viral titers of both rabbit HEV and swine HEV-3, used to challenge rabbits, were adjusted to 10^6^ genomic equivalent (GE) copy number/mL in 4% bovine albumin solution.

### 2.2. RNA Extraction from Fecal and Serum Samples

Serum and fecal samples were collected from all rabbits every week for 8 weeks post infection (wpi). Serum samples were separated from the whole blood by centrifugation at 3000× *g* for 15 min. Fecal samples were suspended in PBS (ratio 1:10), centrifuged at 3000× *g* for 30 min, and the supernatants were collected. The supernatants were re-centrifuged at 13,000× *g* for 10 min and the second supernatants were collected. All serum and fecal samples were stored at −70 °C until use. Viral RNA was extracted from 150 µL of serum and fecal samples using a Patho Gene-spin DNA/RNA kit according to manufacturer’s instructions (Intron, Gyeonggi, Korea). The extracted RNA samples were used for cDNA synthesis or stored at −70 °C.

### 2.3. Detection of Partial HEV Genomic Sequences

Nested RT-PCR with detection limits of 10^2^–10^3^ GE copies was performed to detect partial genomic sequence of the HEV ORF2 with gene-specific primers under conditions as described previously [25]. The correct sizes of the final PCR products were identified by gel electrophoresis. The amplified viral DNA was extracted from the PCR products and cloned into a TA Cloning Vector (RBC™, New Taipei City, Taiwan). The cloned amplicons were transformed into HIT Competent Cells™-DH5α (RBC™). DNA sequences were determined using plasmid DNA extracted from the colony.

### 2.4. ELISA for Anti-HEV Antibodies

Anti-HEV antibody titers were determined in serum samples collected from each of the rabbits in the three groups over 8 weeks using a commercial ELISA kit (Wantai, Beijing, China) according to the manufacturer’s instructions. 

### 2.5. Alanine Aminotransferase (ALT) and Aspartate Aminotransferase (AST) Levels

ALT and AST levels in serum samples collected from each of the rabbits in the three experimental groups over 8 weeks were measured by a UV-assay, according to the procedure by International Federation of Clinical Chemistry and Laboratory Medicine (Neodin, Seoul, Korea), without pyridoxal phosphate activation.

### 2.6. Cytokine Levels

The levels of IL-1, IL-6 and TNF-α in serum samples collected at 0, 2, 3, 5, 6 and 8 wpi were measured by using ELISA kits (MyBioSource, San Diego, CA, USA). IFN-α levels in serum samples collected at 2, 3, 5 and 6 wpi were determined with ELISA kit (MyBioSource). All experimental procedures were conducted according to manufacturer’s instructions.

### 2.7. Histopathology

Liver tissues taken at 8 wpi were fixed using 10% neutral buffered formalin and embedded with paraffin. Tissues were stained with hematoxylin and eosin (H&E) to identify inflammatory lesions, and with Masson’s trichrome for identifying fibrosis. Degree of hepatic inflammation was graded by modification of the Brunt system [27,28]. Scores 0, 1, 2 and 3 represent absence of inflammation, 1 to 2 focal inflammation, 3 to 4 focal inflammation, and more than 4 focal inflammation per 10× objective, respectively.

## 3. Results

### 3.1. Detection of HEV RNA 

The HEV RNA was determined using serum and fecal samples collected over 8 weeks by nested RT-PCR. As expected, no HEV RNA was detected from negative control rabbits during the experimental period (Table 1). 

HEV RNA was detected from the fecal or serum samples collected from all four rabbits (100%) infected with rabbit HEV from 3 to 8 wpi (Table 1). In contrast, HEV RNA was detected from only four of the six rabbits (66.7%) infected with swine HEV-3 from 4 to 7 wpi (Table 1). These results indicate that swine HEV-3 can be transmitted to rabbits across species. However, the infectivity of swine HEV-3 in rabbits was relatively lower than that of rabbit HEV.

### 3.2. Determination of Anti-HEV Antibody Levels

The presence of anti-HEV antibodies was determined from serum samples from each of the rabbits in the three groups. In negative control rabbits, anti-HEV antibody was not detected, as expected (Figure 1). In contrast, seroconversion was seen in serum samples collected from rabbits infected with rabbit HEV or swine HEV (Figure 1). Anti-HEV antibodies appeared from 4 to 5 wpi in all rabbits infected with rabbit HEV (Table 2). In contrast, anti-HEV antibodies were initially detected from 4 to 6 wpi in four out of six rabbits infected with swine HEV-3 (Table 2). However, at 8 wpi, antibody titers in rabbits infected with rabbit HEV were higher than titers in rabbits infected with swine HEV-3 (Figure 1). These results indicate that swine HEV-3 induces production of anti-HEV antibodies from rabbits.

### 3.3. Determination of Liver Enzyme Concentrations

There was no elevation of ALT nor AST levels in serum of rabbits in the negative control group, as expected (Figure 2). While ALT levels in rabbits infected with rabbit HEV or swine HEV seemed to increase at the end of experiments, the concentrations of both ALT and AST in serum samples collected from rabbits infected with either rabbit HEV or swine HEV-3 were not significantly higher than that in negative control rabbits (Figure 2). These results show that infection with rabbit HEV or swine HEV does not induce severe inflammation in the liver tissues of rabbits.

### 3.4. Determination of Cytokine Levels

The levels of proinflammatory cytokines IL-1β, IL-6 and IFN-α were measured in rabbit serum samples collected at 0, 2, 3, 5, 6 and 8 wpi. IFN-α levels were determined in serum samples collected at 2, 3, 5 and 6 wpi. Serum IL-1β, IL-6, TNF, and IFN-α concentrations in rabbits infected with rabbit HEV or swine HEV were not significantly different from serum concentration in negative control rabbits (Appendix A). These results indicate that rabbit HEV and swine HEV do not induce acute systemic inflammation in rabbits.

### 3.5. Histopathology

No inflammatory lesions were seen in the liver tissues of rabbits in the negative control group (Figure 3A). Conversely, histopathological lesions were detected in the liver tissues of rabbits infected with rabbit HEV (Figure 3B). 

Multifocal and diffuse accumulations of inflammatory cells around the central veins, portal areas, and parenchyma were found in rabbits infected with rabbit HEV. Inflammatory cell infiltrates were mostly monocytes. Inflammatory lesions were also seen in liver tissue from rabbits infected with swine HEV-3 (Figure 3C). However, these lesions were less severe than those seen in rabbits infected with rabbit HEV. The average histopathological severity scores of liver tissue from rabbits in the negative control group, the rabbit HEV-infected group, and the swine HEV-3-infected group were 0.25, 2.0 and 1.5, respectively (Table 1). Masson’s trichrome staining of liver tissues demonstrated occasional fibrotic lesions in liver tissue from negative control group rabbits (Figure 3D). In contrast, severe fibrosis was detected in the liver tissues of rabbits infected with rabbit HEV (Figure 3E). Similarly, considerable fibrosis was seen in the liver tissues of rabbits infected with swine HEV-3 (Figure 3F).

## 4. Discussion

Several animal models have been used to understand the pathogenesis and immune responses of HEV. Generally, non-human primates such as chimpanzee, cynomolgus, and rhesus monkeys, are considered the best animals for HEV infection studies because these animals can be infected with various HEV genotypes and produce a similar disease to that observed in humans [16,19,29]. Pigs can also be successfully infected with human strains of HEV-3 and HEV-4, and with rabbit HEV [30,31,32,33]. However, more convenient animal models for the study of HEV cross-species transmission are required. Rabbits can be infected with wild boar HEV-3 and swine HEV-4 [34,35,36]. However, some argue that cross-species transmission of swine HEV-4 and rabbit HEV does not happen between rabbits and pigs that are reared in animal farms in the same regions [37,38,39]. Rabbit HEV strains are known to form a new clade in the HEV-3 strains [1]. Therefore, we anticipated that swine HEV-3 would efficiently infect rabbits.

In this study, we demonstrated that about 67% of rabbits could be infected with swine HEV-3. Swine HEV-3 induced viremia, fecal shedding and seroconversion in virus-infected rabbits. Seroconversion patterns identified in rabbits infected with swine HEV-3 are similar to those in rabbits infected with the wild boar HEV-3 strain [36]. However, fecal viral shedding patterns between the two studies were different. In this study, fecal viral shedding was detected at 5–7 wpi in rabbits infected with swine HEV-3, whereas rabbits infected with wild boar HEV-3 start fecal viral shedding at as early as 3 dpi and continue to 14 dpi [36]. Rabbits inoculated with human HEV-3 showed poor infectivity with only seroconversion, but not viral shedding [23]. Other studies demonstrated that rabbits infected with human and swine HEV-4 strains could induce viral shedding and seroconversion, similar to the results presented here [23,24,34].

Elevated hepatic enzymes indicate inflammation and damage to liver tissues. It is known that the normal physiological ranges of ALT and AST are 55–260 U/L and 10–98 U/L in rabbits, respectively [40]. In this study, liver enzymes ALT and AST were not significantly elevated in rabbits infected with swine HEV-3 or rabbit HEV until 8 wpi. Their ALT and AST levels remained in the normal ranges. However, elevation of ALT was observed in rabbits infected with both human and swine HEV-4 strains [23,24,34]. These discrepancies could be attributed to differences in the virulence and dose of viruses used in the different studies.

Induction of proinflammatory cytokines such as IL-1β, IL-6, IFN-α, and TNF-α, is indicative of acute viral infection in animals [41]. However, rabbits infected with swine HEV-3 or rabbit HEV did not produce high levels of pro-inflammatory cytokines in this study. These results suggest that both rabbit HEV and swine HEV do not induce acute infections in rabbits. However, histopathological examination indicated that rabbit HEV and swine HEV-3 induce mild inflammatory lesions in the liver tissues of infected rabbits, with mononuclear cells predominantly shown to have infiltrated liver tissues. Furthermore, our study demonstrated that rabbit HEV and swine HEV-3 cause fibrosis in the livers of infected rabbits. Some discrepancy observed between histopathological scores and viremia and viral shedding in HEV-infected rabbits might be attributed to the physiological and immunological differences of individual rabbits. According to another study, hepatocellular necrosis is induced in rabbits infected with rabbit HEV [24]. Additionally, multifocal lymphocytic infiltration is known to occur in the liver of rabbits infected with human HEV [23,24]. Severe histopathological signs are associated with high titers of both human and rabbit HEV [23]. Inflammatory lesions with multifocal infiltration of lymphocytes, plasma cells, macrophages, and plasma cells have been observed in pigs infected with swine HEV-3 [13]. A histopathological study of HEV infection has also been conducted in gerbils, which showed lymphocyte and eosinophil infiltration and histiocytic hepatitis [42]. In humans, portal inflammation and interface hepatitis are observed in acute HEV patients [43,44]. However, fibrosis is increased in patients with chronic HEV infection [44,45]. The histopathological findings in our study indicate that mild inflammation and fibrosis are reproducible in the liver tissues of rabbits infected with swine HEV or rabbit HEV.

## 5. Conclusions

Collectively, our results show cross-species transmission in rabbits infected with swine HEV-3. Viremia, fecal viral shedding, and seroconversion were observed, while increases in hepatic enzymes and proinflammatory cytokines did not occur in rabbits infected with swine HEV. Mild hepatic lesions and fibrosis were seen in the liver tissues of rabbits infected with swine HEV. These results indicate that swine HEV-3 can cross the species barrier to infect rabbits where it induces mild hepatitis without severe inflammation.

## Figures and Tables

**Figure 1 viruses-12-00053-f001:**
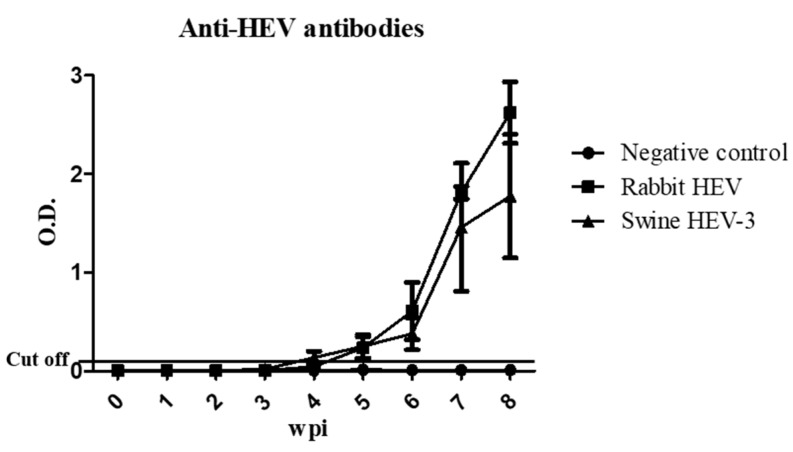
Determination of anti-HEV antibody titers. Anti-HEV antibody levels in serum samples were measured by ELISA at 0–8 wpi. Anti-HEV antibody levels were elevated in rabbits infected with rabbit HEV and swine HEV-3.

**Figure 2 viruses-12-00053-f002:**
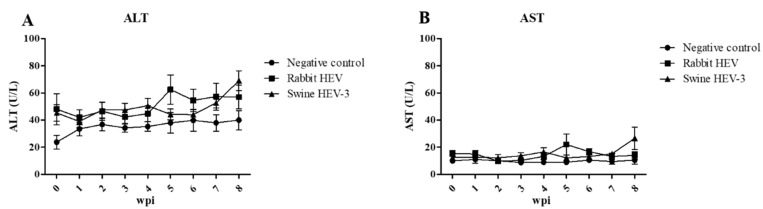
Determination of hepatic enzyme levels. (**A**) ALT and (**B**) AST levels measured by UV-assay at 0–8 wpi in HEV-infected rabbits versus non-infected control rabbits.

**Figure 3 viruses-12-00053-f003:**
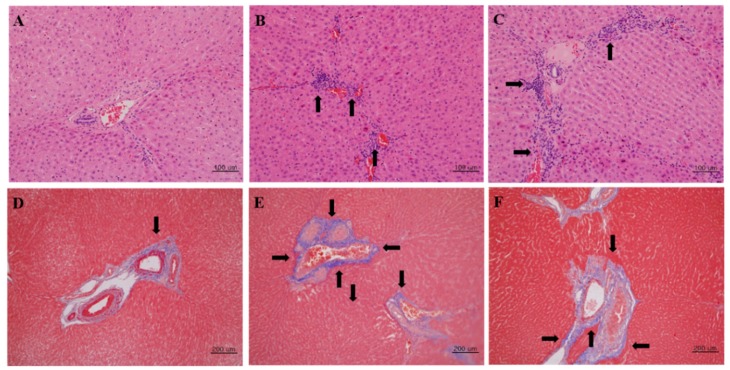
Detection of histopathological lesions in the liver tissues. Liver tissues collected at 8 wpi from (**A**) negative control, (**B**) rabbit HEV-infected, and (**C**) swine HEV-3-infected rabbits were stained using H&E. Liver tissues of (**D**) negative control, (**E**) rabbit HEV-infected, and (**F**) swine HEV-3-infected rabbits were stained using Masson’s trichrome staining method. Arrows indicate (**A**–**C**) inflammation and (**D**–**F**) fibrosis.

**Table 1 viruses-12-00053-t001:** Detection of HEV in serum and fecal samples and evaluation of histopathological lesions in the livers of rabbits.

Group	wpi	1	2	3	4	5	6	7	8	Histopathological Score	Average of Scores
	No.	F/S *	F/S	F/S	F/S	F/S	F/S	F/S	F/S		
Negative Control	1	−/−	−/−	−/−	−/−	−/−	−/−	−/−	−/−	0	0.25
	2	−/−	−/−	−/−	−/−	−/−	−/−	−/−	−/−	0	
	3	−/−	−/−	−/−	−/−	−/−	−/−	−/−	−/−	1	
	4	−/−	−/−	−/−	−/−	−/−	−/−	−/−	−/−	0	
Rabbit HEV	5	−/−	−/−	−/−	−/−	−/+	−/+	−/−	−/−	1	2.0
	6	−/−	−/−	+/+	+/+	+/+	+/+	−/−	−/−	2	
	7	−/−	−/−	−/−	−/−	−/−	+/−	+/−	+/−	3	
	8	−/−	−/−	−/−	−/−	+/−	+/−	−/−	−/−	2	
Swine HEV-3	9	−/−	−/−	−/−	−/−	−/−	−/−	−/−	−/−	1	1.5
	10	−/−	−/−	−/−	−/−	−/+	+/−	−/−	−/−	1	
	11	−/−	−/−	−/−	−/−	−/−	−/−	−/−	−/−	0	
	12	−/−	−/−	−/−	−/+	+/+	+/+	+/−	−/−	3	
	13	−/−	−/−	−/−	−/−	−/−	−/+	−/−	−/−	1	
	14	−/−	−/−	−/−	−/−	+/−	−/−	−/−	−/−	3	

* F: fecal sample, S: serum sample.

**Table 2 viruses-12-00053-t002:** Detection of anti-HEV antibodies from each rabbit.

Group	wpi	1	2	3	4	5	6	7	8
	No.	Serum	Serum	Serum	Serum	Serum	Serum	Serum	Serum
Negative Control	1	−	−	−	−	−	−	−	−
	2	−	−	−	−	−	−	−	−
	3	−	−	−	−	−	−	−	−
	4	−	−	−	−	−	−	−	−
Rabbit HEV	5	−	−	−	−	+	+	+	+
	6	−	−	−	+	+	+	+	+
	7	−	−	−	+	+	+	+	+
	8	−	−	−	+	+	+	+	+
Swine HEV−3	9	−	−	−	−	−	−	−	−
	10	−	−	−	−	+	+	+	+
	11	−	−	−	−	−	−	−	−
	12	−	−	−	+	+	+	+	+
	13	−	−	−	−	−	+	+	+
	14	−	−	−	+	+	+	+	+

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
