# Peer review of "Cross-Species Transmission of Swine Hepatitis E Virus Genotype 3 to Rabbits"

_viruses, 2020, doi:10.3390/v12010053_

Round 1

Reviewer 1 Report

Han et al present an interesting paper about the infectivity of swine derived HEV strain in rabbit models. The findings suggest that, despite swine derived strain has a lower infectivity than HEV-3ra, could be used in rabbits for in vivo studies. The paper is well written, but I have several comments that need to be addressed by authors, overall regarding to Methods:

Several important points of the methodology are lacking.

- How animals were infected (orally, intravenous inoculation, intraperitoneal…? This point is major in sense to interpret the results, taking in mind the lower infection rate in the HEV-3ra group.

- The detection limit of the PCR assay is mandatory. This should be given for both serum and fecal samples.

Authors response in this sense should be also included in discussion. The reason is that is striking that really few animals were positive in serum (including in the positive control group).

Results.

- Figure 1 should include variability range (graphical representation).

- I´m confusing with the expression “HEV RNA titers” in the present study do to, in my knowledge, authors do not perform qPCR. Please clarify this point.  

- Lines 123 to 127 are redundant with lines 110 to 114. Please, correct.

- How authors could explain that the cytokines profile and ALT/AST are similar between experimental and controls (positive and mock) groups but not histopathological examination? I presume that any marker should be altered.

Conclusions

- The swine HEV strain used in the study, has a BLAST score of 96.91% with a HEV viral strain isolated from a patient in Japan (AB291956). Due to it has been stated by others that rabbits are not susceptible to human derived HEV-3, I considered that this point should be discussed in sense to increase the value of the present work.   

Minor comments in the introduction section:

- Please consider using Orthohepeviridae rather than Herpesviridae (in fact is Hepeviridae and not Herpesviridae).

- In lines 32 to 35 are mixing concept of genotypes 1 and 2 (high mortality during pregnancy) and genotype 3 (chronic hepatitis). I recommend a well differentiation in sense to avoid confusion by a potential readers.   

Author Response

Dear Reviewers

We did our best to improve our manuscript by changing the contents following your comments and suggestions. We highlighted the changed parts as red colors. We appreciate your kind suggestions and helps. Here are our responses to your comments.

COMMENTS TO THE AUTHOR:

Reviewer #1:  Han et al present an interesting paper about the infectivity of swine derived HEV strain in rabbit models. The findings suggest that, despite swine derived strain has a lower infectivity than HEV-3ra, could be used in rabbits for in vivo studies. The paper is well written, but I have several comments that need to be addressed by authors, overall regarding to Methods:

Several important points of the methodology are lacking.

Q1) How animals were infected (orally, intravenous inoculation, intraperitoneal…? This point is major in sense to interpret the results, taking in mind the lower infection rate in the HEV-3ra group.

Response: page 4, line 16. We described how rabbits were infected with rabbit HEV and swine HEV: “Rabbits were intravenously infected with rabbit HEV and swine HEV-3”

Q2) The detection limit of the PCR assay is mandatory. This should be given for both serum and fecal samples.

Response: page 5, line 10. We described the detection limits of this PCR as 102-103 genomic equivalent copies.

Q3) Authors response in this sense should be also included in discussion. The reason is that is striking that really few animals were positive in serum (including in the positive control group).

Response: page 14, line 5. We fully understand your opinion. But whenever we did infection experiments with HEV or other virus, we frequently found relatively low infectivity of the virus in experimental animals. Our results indicated that every rabbit infected with rabbit HEV demonstrated HEV in serum or fecal sample. In addition, HEV could be detected from serum or fecal samples in about 67% of rabbits which were infected with swine HEV-3. We do not think that low positive results in rabbits infected with rabbit HEV and swine HEV would be the low detection limits of the PCR we employed in this study. Please understand our opion.

Results.

Q4) Figure 1 should include variability range (graphical representation).

Response: page 9. We changed figure 1 which shows variability ranges.

Q5) I´m confusing with the expression “HEV RNA titers” in the present study do to, in my knowledge, authors do not perform qPCR. Please clarify this point.  

Response: page 6, line 21. We deleted the term “titers” and described as “HEV RNA”. Your opinion is right, we did not perform qPCR.

Q6) Lines 123 to 127 are redundant with lines 110 to 114. Please, correct.

Response: We removed the lines 123 to 127. We think the redundant sentences might be copied during insertion of figures, sorry about that.

Q7) How authors could explain that the cytokines profile and ALT/AST are similar between experimental and controls (positive and mock) groups but not histopathological examination? I presume that any marker should be altered.

Response: pages 15, lines 22-23; page 16, line 1. We carefully re-evaluated our results, but there was no wrongly expressed data. Therefore, we could reach the conclusion like this: “swine HEV-3 can cross the species barrier to infect rabbits where it induces mild hepatitis without severe inflammation”. Please, acknowledge our data.

Conclusions

Q8) The swine HEV strain used in the study, has a BLAST score of 96.91% with a HEV viral strain isolated from a patient in Japan (AB291956). Due to it has been stated by others that rabbits are not susceptible to human derived HEV-3, I considered that this point should be discussed in sense to increase the value of the present work.   

Response: page 14, lines 11-14. We indicated poor infection of human HEV-3 in rabbits and compared our results and those determined in rabbits infected with swine HEV-4.

Other minor comments:

Q9) Please consider using Orthohepeviridae rather than Herpesviridae (in fact is Hepeviridae and not Herpesviridae).

Response: page 3, line 3. We corrected the term like this “Hepeviridae

Q10) In lines 32 to 35 are mixing concept of genotypes 1 and 2 (high mortality during pregnancy) and genotype 3 (chronic hepatitis). I recommend a well differentiation in sense to avoid confusion by a potential readers.   

Response: page 3, lines 8-10. We corrected the confusing description more clearly as you suggested.

Reviewer #2: Han et al. interested in the possibility of infection of rabbits with HEV3 strains isolated from pigs. The question is interesting but the experiments do not support clearly the conclusions. A more detailed protocol of the infection is needed to better interpret the results and additional experiments are required to support the conclusions.

Major comments

Q1) Lines 59-68: How were the rabbits infected? Intra hepatic injection, oral route? Please clarify. This is a key point to interpret the data.

Response: page 4, line 16. We described how rabbits were infected with rabbit HEV and swine HEV: “Rabbits were intravenously infected with rabbit HEV and swine HEV-3”

Q2) Lines 100-102: could you detail a bit more the histopathological severity score? Rabbit 3 in the negative control group has a score of 1, and 2 rabbits infected with the swine HEV3 strain (rabbits n°10 and 12) have also a score of 1. The 2 rabbits (n° 13 and 14) with a score of 3 have only transient viremia or secretion, but severe lesions. Conversely, rabbit n°3 was viremic for 3 weeks but present a score of 1. How can this observation be interpreted?

Response: page 6, lines 15-17. We described more detailed methodology which shows scoring system. The histological examination was conducted by a pathologist Dr. YK Choi in our department. We fully understand your opinion about inconsistency of the histological results, viremia, and viral shedding in individual rabbits. We think the histopathological score 1 from one negative control rabbit might be induced by unknown reason during experimental period, such as physiological conditions, diet, non-pathological agent, and other factors. Some discrepancy observed between histopathological scores and viremia and viral shedding in HEV-infected rabbits might be attributed to the physiological and immunological differences of individual rabbits. We think these kinds of a little bit inconsistent results are sometimes encountered in animal experiments. Please, understand our situations and opinions.

Q3) Figure 3: when were the liver samples collected?

Response: page 6, line 13; page 13, line 3. We indicated when the live samples were collected in the text and figure legend.                                                      

Q4) Line 132: did the antibodies appeared in all rabbits of the 2 infected groups? Or were they detected only in viremic rabbits? I suggest making a table similar to the table 1. In addition, the detection of HEV antibodies following swine HEV3 infection is not sufficient to claim that swine HEV3 replicated in rabbits (line 135).

Response: page 9, lines 6-11; page 10. We added Table 2 in the manuscritp as your suggestion. We clearly described seroconversion rates from rabbits. We also changed the previous last sentence of the section like this: “These results indicate that swine HEV-3 can infect rabbits and induce production of anti-HEV antibodies from them”

Q5) Line 142-146: what is the “physiologic” range for ALT in rabbits? Did the ALT activity only increase in viremic rabbits infected either with HEV3 swine or HEV3 rabbit strains? Please specify in the text. In addition, the increase of ALT concentration at week 8 in the group infected with HEV3 swine is surprising since at that time, no rabbit was viremic.

Response: page 14, lines 15-18. We described the normal ranges of ALT and AST in the text by referring a reference [41]. We also indicated that all rabbits infected with rabbit HEV and swine HEV-3 remained in normal ranges of both enzymes.

Q6) Line 148-150: can the levels of pro-inflammatory cytokines be determined at weeks 5 and 6? It would be more consistent since most animals are viremic at that time.

Response: page 11, lines 14-15; page 12. We examined the levels of pro-inflammatory cytokines at 5 and 6 wpi and presented a new supplementary figure 1 as you suggested. But we could not find elevation of any cytokine during the experimental period.

Q7) Could a PCR be performed to detect the negative strand of HEV RNA in liver biopsies?

Response: Unfortunately, we did not perform PCR for detection of HEV negative RNA in the liver biopsy. We immediately fixed and embedded the liver samples after euthanizing of rabbit for histopathological examination. That was our mistake. Please, understand our situation.

Other minor comments:

Q8) Line 12 and 29: HEV is a quasi-enveloped virus, with dual form: associated with lipids (but without peplomer) in the blood but naked in the stools.

Response: page 2, line 1 and page 3, line 3. We changed the term “a non-enveloped virus” to “a quasi-enveloped virus” as you suggested.

Q9) Line 29 : HEV belongs to the Hepeviridae family, not the Herpesviridae

Response: page 3, line 3. Sorry, it was our mistake and we corrected it to “Hepeviridae

Q10) Line 47: the rabbit HEV strain was also detected in humans, for instance Sahli R et al., J Hepatol. 2019 or Abravanel et al., Emerg Infect Dis. 2017.

Response: page 3, line 22. We indicated that rabbit HEV could infect humans. We also added two references you suggested (Sahli R et al., J Hepatol. 2019 and Abravanel et al., Emerg Infect Dis. 2017)

Q11) Line 66: HEV exists in a dual form: quasi enveloped in the blood and supernatant, but naked in the stools. It would be better to specify that HEV from stools was used to avoid the reader to look for this information in the 2 cited papers 23-24.

Response: page 4, lines 17-18. We indicated that rabbit and swine HEV were obtained from stool samples as you suggested.

Q12) Lines 110-114 and lines 123-217 are the same, so are the titles 3.2 and 3.3.

Response: page 9, line 2 and page 11, line 1. We correctly changed the titles of 3.2 and 3.3.

Reviewer 2 Report

Han et al. interested in the possibility of infection of rabbits with HEV3 strains isolated from pigs. The question is interesting but the experiments do not support clearly the conclusions. A more detailed protocol of the infection is needed to better interpret the results and additional experiments are required to support the conclusions.

Major concerns:

Lines 59-68: How were the rabbits infected? Intra hepatic injection, oral route? Please clarify. This is a key point to interpret the data.

Lines 100-102: could you detail a bit more the histopathological severity score? Rabbit 3 in the negative control group has a score of 1, and 2 rabbits infected with the swine HEV3 strain (rabbits n°10 and 12) have also a score of 1. The 2 rabbits (n° 13 and 14) with a score of 3 have only transient viremia or secretion, but severe lesions. Conversely, rabbit n°3 was viremic for 3 weeks but present a score of 1. How can this observation be interpreted?

Figure 3: when were the liver samples collected?

Line 132: did the antibodies appeared in all rabbits of the 2 infected groups? Or were they detected only in viremic rabbits? I suggest to make a table similar to the table 1. In addition, the detection of HEV antibodies following swine HEV3 infection is not sufficient to claim that swine HEV3 replicated in rabbits (line 135).

Line 142-146: what is the “physiologic” range for ALT in rabbits? Did the ALT activity only increase in viremic rabbits infected either with HEV3 swine or HEV3 rabbit strains? Please specify in the text. In addition, the increase of ALT concentration at week 8 in the group infected with HEV3 swine is surprising since at that time, no rabbit was viremic.

Line 148-150: can the levels of pro-inflammatory cytokines be determined at weeks 5 and 6? It would be more consistent since most animals are viremic at that time.

Could a PCR be performed to detect the negative strand of HEV RNA in liver biopsies?

Minor concerns:

Line 12 and 29: HEV is a quasi-enveloped virus, with dual form: associated with lipids (but without peplomer) in the blood but naked in the stools.

Line 29 : HEV belongs to the Hepeviridae family, not the Herpesviridae

Line 47: the rabbit HEV strain was also detected in humans, for instance Sahli R et al., J Hepatol. 2019 or Abravanel et al., Emerg Infect Dis. 2017.

Line 66: HEV exists in a dual form: quasi enveloped in the blood and supernatant, but naked in the stools. It would be better to specify that HEV from stools was used to avoid the reader to look for this information in the 2 cited papers 23-24.

Lines 110-114 and lines 123-217 are the same, so are the titles 3.2 and 3.3.

Author Response

Dear Reviewers

We did our best to improve our manuscript by changing the contents following your comments and suggestions. We highlighted the changed parts as red colors. We appreciate your kind suggestions and helps. Here are our responses to your comments.

Major comments

Q1) Lines 59-68: How were the rabbits infected? Intra hepatic injection, oral route? Please clarify. This is a key point to interpret the data.

Response: page 4, line 16. We described how rabbits were infected with rabbit HEV and swine HEV: “Rabbits were intravenously infected with rabbit HEV and swine HEV-3”

Q2) Lines 100-102: could you detail a bit more the histopathological severity score? Rabbit 3 in the negative control group has a score of 1, and 2 rabbits infected with the swine HEV3 strain (rabbits n°10 and 12) have also a score of 1. The 2 rabbits (n° 13 and 14) with a score of 3 have only transient viremia or secretion, but severe lesions. Conversely, rabbit n°3 was viremic for 3 weeks but present a score of 1. How can this observation be interpreted?

Response: page 6, lines 15-17. We described more detailed methodology which shows scoring system. The histological examination was conducted by a pathologist Dr. YK Choi in our department. We fully understand your opinion about inconsistency of the histological results, viremia, and viral shedding in individual rabbits. We think the histopathological score 1 from one negative control rabbit might be induced by unknown reason during experimental period, such as physiological conditions, diet, non-pathological agent, and other factors. Some discrepancy observed between histopathological scores and viremia and viral shedding in HEV-infected rabbits might be attributed to the physiological and immunological differences of individual rabbits. We think these kinds of a little bit inconsistent results are sometimes encountered in animal experiments. Please, understand our situations and opinions.

Q3) Figure 3: when were the liver samples collected?

Response: page 6, line 13; page 13, line 3. We indicated when the live samples were collected in the text and figure legend.                                                      

Q4) Line 132: did the antibodies appeared in all rabbits of the 2 infected groups? Or were they detected only in viremic rabbits? I suggest making a table similar to the table 1. In addition, the detection of HEV antibodies following swine HEV3 infection is not sufficient to claim that swine HEV3 replicated in rabbits (line 135).

Response: page 9, lines 6-11; page 10. We added Table 2 in the manuscritp as your suggestion. We clearly described seroconversion rates from rabbits. We also changed the previous last sentence of the section like this: “These results indicate that swine HEV-3 can infect rabbits and induce production of anti-HEV antibodies from them”

Q5) Line 142-146: what is the “physiologic” range for ALT in rabbits? Did the ALT activity only increase in viremic rabbits infected either with HEV3 swine or HEV3 rabbit strains? Please specify in the text. In addition, the increase of ALT concentration at week 8 in the group infected with HEV3 swine is surprising since at that time, no rabbit was viremic.

Response: page 14, lines 15-18. We described the normal ranges of ALT and AST in the text by referring a reference [41]. We also indicated that all rabbits infected with rabbit HEV and swine HEV-3 remained in normal ranges of both enzymes.

Q6) Line 148-150: can the levels of pro-inflammatory cytokines be determined at weeks 5 and 6? It would be more consistent since most animals are viremic at that time.

Response: page 11, lines 14-15; page 12. We examined the levels of pro-inflammatory cytokines at 5 and 6 wpi and presented a new supplementary figure 1 as you suggested. But we could not find elevation of any cytokine during the experimental period.

Q7) Could a PCR be performed to detect the negative strand of HEV RNA in liver biopsies?

Response: Unfortunately, we did not perform PCR for detection of HEV negative RNA in the liver biopsy. We immediately fixed and embedded the liver samples after euthanizing of rabbit for histopathological examination. That was our mistake. Please, understand our situation.

Other minor comments:

Q8) Line 12 and 29: HEV is a quasi-enveloped virus, with dual form: associated with lipids (but without peplomer) in the blood but naked in the stools.

Response: page 2, line 1 and page 3, line 3. We changed the term “a non-enveloped virus” to “a quasi-enveloped virus” as you suggested.

Q9) Line 29 : HEV belongs to the Hepeviridae family, not the Herpesviridae

Response: page 3, line 3. Sorry, it was our mistake and we corrected it to “Hepeviridae

Q10) Line 47: the rabbit HEV strain was also detected in humans, for instance Sahli R et al., J Hepatol. 2019 or Abravanel et al., Emerg Infect Dis. 2017.

Response: page 3, line 22. We indicated that rabbit HEV could infect humans. We also added two references you suggested (Sahli R et al., J Hepatol. 2019 and Abravanel et al., Emerg Infect Dis. 2017)

Q11) Line 66: HEV exists in a dual form: quasi enveloped in the blood and supernatant, but naked in the stools. It would be better to specify that HEV from stools was used to avoid the reader to look for this information in the 2 cited papers 23-24.

Response: page 4, lines 17-18. We indicated that rabbit and swine HEV were obtained from stool samples as you suggested.

Q12) Lines 110-114 and lines 123-217 are the same, so are the titles 3.2 and 3.3.

Response: page 9, line 2 and page 11, line 1. We correctly changed the titles of 3.2 and 3.3.

Round 2

Reviewer 2 Report

I thank the authors for their answers. I have 2 minor points.

Page 9 lines 10-11: Again, appearance of antibodies does not mean infection. I would write “these results indicate that swine-HEV3 induce production of anti-HEV antibodies from rabbits”.

Part of response to Q2 could be added to the discussion to help the reader to understand discrepancies: “Some discrepancy observed between histopathological scores and viremia and viral shedding in HEV-infected rabbits might be attributed to the physiological and immunological differences of individual rabbits.”

Author Response

Reviewer’ comments (Revision 2)

We modified two minor parts as you suggested in our revised manuscript. We indicated the changed parts as blue letters in the text. Thank you so much for your kind help to improve our manuscript.

I thank the authors for their answers. I have 2 minor points.

Page 9 lines 10-11: Again, appearance of antibodies does not mean infection. I would write “these results indicate that swine-HEV3 induce production of anti-HEV antibodies from rabbits”.

Response: page 9, lines 9-10. We modified the previous sentence like this “These results indicate that swine-HEV3 induce production of anti-HEV antibodies from rabbits” as you suggested.

Part of response to Q2 could be added to the discussion to help the reader to understand discrepancies: “Some discrepancy observed between histopathological scores and viremia and viral shedding in HEV-infected rabbits might be attributed to the physiological and immunological differences of individual rabbits.”

Response: page 15, lines 6-8. We added the sentence “Some discrepancy observed between histopathological scores and viremia and viral shedding in HEV-infected rabbits might be attributed to the physiological and immunological differences of individual rabbits” as you suggested.